# Supplemental Ferulic Acid Inhibits Total Body Irradiation-Mediated Bone Marrow Damage, Bone Mass Loss, Stem Cell Senescence, and Hematopoietic Defect in Mice by Enhancing Antioxidant Defense Systems

**DOI:** 10.3390/antiox10081209

**Published:** 2021-07-28

**Authors:** Sajeev Wagle, Hyun-Jaung Sim, Govinda Bhattarai, Ki-Choon Choi, Sung-Ho Kook, Jeong-Chae Lee, Young-Mi Jeon

**Affiliations:** 1Cluster for Craniofacial Development and Regeneration Research, Institute of Oral Bioscience, Jeonbuk National University School of Dentistry, Jeonju 54896, Korea; swagle@jbnu.ac.kr (S.W.); skynow@hanmail.net (H.-J.S.); govinda@jbnu.ac.kr (G.B.); 2Department of Bioactive Material Sciences and Research Center of Bioactive Materials, Jeonbuk National University, Jeonju 54896, Korea; 3Grassland and Forages Research Center, National Institute of Animal Science, Cheonan 31002, Korea; choiwh@korea.kr

**Keywords:** ferulic acid, total body irradiation, bone marrow microenvironment, stem cell senescence, bone marrow injury, reactive oxygen species, antioxidant defense system

## Abstract

While total body irradiation (TBI) is an everlasting curative therapy, the irradiation can cause long-term bone marrow (BM) injuries, along with senescence of hematopoietic stem cells (HSCs) and mesenchymal stem cells (MSCs) via reactive oxygen species (ROS)-induced oxidative damages. Thus, ameliorating or preventing ROS accumulation and oxidative stress is necessary for TBI-requiring clinical treatments. Here, we explored whether administration of ferulic acid, a dietary antioxidant, protects against TBI-mediated systemic damages, and examined the possible mechanisms therein. Sublethal TBI (5 Gy) decreased body growth, lifespan, and production of circulating blood cells in mice, together with ROS accumulation, and senescence induction of BM-conserved HSCs and MSCs. TBI also impaired BM microenvironment and bone mass accrual, which was accompanied by downregulated osteogenesis and by osteoclastogenic and adipogenic activation in BM. Long-term intraperitoneal injection of ferulic acid (50 mg/kg body weight, once per day for 37 consecutive days) protected mice from TBI-mediated mortality, stem cell senescence, and bone mass loss by restoring TBI-stimulated disorders in osteogenic, osteoclastic, and adipogenic activation in BM. In vitro experiments using BM stromal cells supported radioprotective effects of ferulic acid on TBI-mediated defects in proliferation and osteogenic differentiation. Overall, treatment with ferulic acid prevented TBI-mediated liver damage and enhanced endogenous antioxidant defense systems in the liver and BM. Collectively, these results support an efficient protection of TBI-mediated systemic defects by supplemental ferulic acid, indicating its clinical usefulness for TBI-required patients.

## 1. Introduction

Radiotherapy via localized or total body irradiation (TBI), in combination with surgical operation, is a common treatment for cancer patients. TBI in moderate or high doses is also applied in bone marrow transplantation therapy. However, TBI may impair the bone marrow (BM) microenvironment, hematopoietic development, and stem cell functions, eventually causing irrecoverable systemic damages [1,2]. TBI-mediated damages are closely associated with abnormal accumulation of reactive oxygen species (ROS) in BM and BM-conserved stem cells, such as hematopoietic stem cells (HSCs) and mesenchymal stem cells (MSCs) [3,4,5]. Persistent and prolonged ROS generation induces stem cell senescence and self-renewal defect in BM, which contributes oxidative-stress-associated long-term BM injuries [3,4,5]. TBI-induced BM injury also disrupts bone mass accrual, and this disruption is associated with the high radiation-absorbing property of bone compared to surrounding soft tissues [6,7].

As TBI-induced adverse events are to be great challenges to cancer patients and in stem cell transplantation therapy, numerous studies have focused on the development of bioactive materials that effectively prevent TBI-mediated systemic impairments. In this regard, many reports indicate a clinical usefulness of naturally occurring hydroxycinnamates, a class of major phenolic compounds, in attenuating and/or protecting TBI-mediated oxidative damages [8,9]. Of the hydroxycinnamates, ferulic acid (4-hydroxy-3-methoxycinamic acid) is a dietary antioxidant that protects radiation-induced damages by tremendously ameliorating cellular ROS accumulation [10,11,12]. This protection is related to its chemical property possessing three distinctive structural motifs that participate in free radical scavenging [13]. A recent report also suggests that supplemental ferulic acid enhances the healing of irradiation-mediated bone defect by maintaining stemness of skeletal stem cells, as well as by activating mitogen-activated protein kinases in the cells [14]. In addition to radioprotection, ferulic acid is also known to exhibit various pharmacological and medicinal activities, including anti-aging, anti-inflammatory, anti-diabetic, anti-cancer, and neuroprotective roles [15].

Given that irradiation induces oxidative-stress-associated cellular DNA damage and senescence of HSCs [16,17], we suggest that cellular mechanisms by which ferulic acid exerts radioprotection are closely associated with its ability to scavenge ROS and to ameliorate TBI-mediated oxidative stress. However, the underlying mechanisms of how ferulic acid protects BM and BM-conserved HSCs against TBI-mediated damages still remain to be defined; the roles of ferulic acid on TBI-mediated impairments in MSC functions and bone mass accrual are not completely understood. Furthermore, the impacts of ferulic acid on TBI-mediated BM injuries, and oxidative systemic disorders by its long-term administration are not investigated. Here, we administered mice with ferulic acid intraperitoneally once per day for 37 consecutive days, from 7 days before and to 30 days after TBI. We explored whether the treatment with ferulic acid protects TBI-induced oxidative damages on HSCs, hematopoietic development, and maintenance of BM microenvironment together with the associated mechanisms. We also investigated how long-term administration of ferulic acid affects the functions of BM-conserved MSCs, bone mass accrual, and life span in TBI-exposed mice. The current findings not only demonstrate radioprotective potentials of ferulic acid in mice and the associated mechanisms, but also provide evidence that ferulic acid is a supplemental antioxidant that ameliorates or prevents TBI-mediated oxidative damages.

## 2. Material and Methods

### 2.1. Chemicals, Laboratory Equipment, and Mice

Ferulic acid was purchased from Sigma-Aldrich Co. LLC (CAS:537-98-4; St. Louis, MI, USA), and 2’,7’-dichlorodihydrofluorescein-diacetate (DCF-DA) was from Abcam (Cambridge, UK). Fetal bovine serum (FBS) was purchased from HyClone Laboratories (Logan, UT, USA). Unless specified otherwise, other chemicals and laboratory consumables were purchased from Sigma-Aldrich Co. LLC and Falcon Labware (BD Biosciences, Franklin Lakes, NJ, USA), respectively. C57BL/6 mice (6 weeks old) were purchased from Damul Science (Daejeon, Republic of Korea) and equilibrated for 7 days before use. During the experimental period, all mice were housed at 22 ± 1 °C and 55 ± 5% humidity, along with 12 h light/dark autocycle, allowing *ad libitum* feeding in the Animal Center of School of Dentistry, Jeonbuk National University.

### 2.2. Ferulic Acid Administration and TBI

Mice were divided into three groups: non-TBI mice (control group), TBI mice with vehicle injection (TBI group), and TBI mice administered with ferulic acid (FA+TBI group). FA+TBI group intraperitonially received ferulic acid (50 mg/kg body weight) once per day for 37 consecutive days, from 7 days before and 30 days after TBI, whereas TBI group was injected with phosphate-buffered saline (PBS; vehicle solution) for the same days. TBI and FA+TBI groups were exposed to 5 Gy TBI with γ-rays by regulating dosage time (0.66 Gy/min) that was based on the radioactive half-life of γ-rays on a rotating platform (Model 109-85 series- JL Shepherd & Associates, San Fernando, CA, USA).

### 2.3. Flowcytometry and Blood Cell Counting

MSCs and HSCs were harvested from femoral and tibial bones of mice by flushing them with PBS using a 5 mL syringe after cutting the ends of the bones. After the removal of red blood cells (RBC), cells were analyzed by multicolor flow cytometry (BD Aria III, BD Biosciences, Franklin Lakes, NJ, USA), and the phenotypical identification of cell populations was performed using FlowJo software (FLOWJO, Ashland, OR, USA) at the Center for University-Wide Research Facilities of Jeonbuk National University. HSCs (CD150^+^CD48^−^LSK) and MSCs (CD29^+^CD105^+^LSK) were characterized with their specific lineage antibodies (antibodies were purchased from BD Biosciences, unless specified otherwise). Briefly, MSCs were characterized by PE-Cy7-conjugated lineage cocktail, APC-Cy7-conjugated anti-Sca-1 (eBioscience, Waltham, MA, USA), PE-conjugated anti-CD29, and APC-conjugated anti-CD105 antibodies, whereas HSCs were characterized by PE-Cy7-conjugated lineage cocktail, FITC- or PE-conjugated anti-Sca-1, APC-conjugated anti-c-Kit, APC-Cy7-conjugated anti-CD48 (eBioscience), and PerCP/Cy5.5-conjugated CD150 (eBioscience) antibodies. The lineage-marker-conjugated MSCs and HSCs were further stained with MitoSox^TM^ Red (Invitrogen, Carlsbad, CA, USA) or Di-β-D-galactopyranoside (C_12_FDG; Molecular probes, Eugene, OR, USA) to analyze levels of cellular ROS accumulation and senescence-associated β-galactosidase (SA-β-gal) activity. Peripheral blood samples were collected from mice via a tail vein cutting or a cardiac puncture into K_2_EDTA-treated tubes (BD Biosciences) 2, 7, 30, and 60 days after TBI. Levels of circulating white blood cells (WBC), granulocytes, lymphocytes, platelets, and RBC were analyzed using an automated blood cell counter (Sysmex XE-2100; TOA Medical Electronics Co., Kobe, Japan).

### 2.4. Micro-Computed Tomography (μCT) Analysis

The femurs were isolated from mice at 60 days post-TBI and scanned using a desktop scanner (1076 Skyscan Micro-CT; Skyscan, Kontich, Belgium), followed by analysis using CTAn software (Skyscan). The X-ray source was set at 75 KV and 100 μA, with a pixel size of 18 mm. The image slices were reconstructed using a cone-beam reconstruction software based on the Feldkamp algorithm (Dataviewer; Skyscan, Belgium). On the stacked reconstruction of μCT cross-section images, manual regions of interest (ROIs) were drawn. The volume of interest (VOI) consisted of a stack of ROIs drawn over 201 cross-sections, resulting in a height of 5 mm extending from the plate of distal femur to trabecular region. Based on the reconstructed three-dimensional (3D) μCT images, values of bone parameters, including bone volume (BV, mm^3^), bone surface (BS, mm^2^), bone specific surface (BS/BV, %), and bone volume percentage (BV/TV, %), were calculated. Bone mineral density (BMD, g/cm^3^) was also determined by converting the attenuation data for VOI into Hounsfield units and BMD units using phantoms (SkyScan) that had a standard density corresponding to mouse bone.

### 2.5. Histological Analyses

Femurs were isolated from mice 30 and 60 days after TBI and fixed with a 4% paraformaldehyde solution for 48 h, followed by decalcification in 10% EDTA at 4 °C for 4 weeks. The tissue, including the trabecular region, was dehydrated in series of alcohol, embedded in paraffin, and sectioned into 5.0 µm in thickness. For hematoxylin and eosin (H & E) staining, tissue sections were treated with hematoxylin solution (Gill No. 3) before counterstaining with 0.25% eosin Y Stain (bioWORLD Life Sciences, Dublin, OH, USA). Parts of tissue sections were subjected to tartrate-resistant acid phosphatase (TRAP) staining using a leukocyte acid phosphatase kit (Cosmo Bio, Tokyo, Japan), followed by counterstaining with hematoxylin. The levels of osteoprotegerin (OPG; BS1862, Bioworld Technology, St. Louis Park, MN, USA), receptor activator of nuclear factor κB ligand (RANKL; ALX-804-243, Enzo Life Sciences, Inc., Farmingdale, NY, USA), osteocalcin (OCN; AB10911, Millipore corporation, Temecula, CA, YSA), runt-related transcription factor 2 (RUNX2; BS2831), cathepsin K (CTSK; SC-48353, Santa Cruz Biotechnology, Inc., Dallas, TX, USA), nuclear factor erythroid 2-related factor 2 (NRF2; BS1258), and adiponectin (ab22554, Abcam) were evaluated by immunohistochemistry (IHC). In IHC assay, tissue sections were stained with each primary antibody (1:200–400 dilutions) specific to the factors, and the expression patterns were determined using rabbit-anti- or mouse-anti-Vectastain ABC DAB-HRP kits (Vector Laboratories, Burlingame, CA, USA). All procedures for TRAP and IHC staining followed the manufacturer’s instructions, and the stained sections were observed under a light microscope (EL-Einsatz 451888, Carl Zeiss, Ostalbkreis, Germany).

### 2.6. Isolation and Culture of BM Stromal Cells (BMSCs)

Whole BM cells were isolated from femurs and tibias of mice at 30 and 60 days post-TBI. Cells were resuspended in alpha-minimum essential media (αMEM, Thermo Fisher Scientific, Waltham, MA, USA) and centrifuged at 2000× *g* for 3 min. The pellets were spread onto 60 mm culture plates and incubated in growth medium (αMEM supplemented with 2 mM glutamine, 100 IU/mL penicillin G, 100 μg/mL streptomycin, and 20% FBS). On the second day, nonadherent cells were removed and, after 12 days of additional incubation, the adherent cells were harvested to use as BMSCs.

### 2.7. Assay for BMSC Proliferation

To evaluate a direct effect of ferulic acid on cell proliferation, BMSCs isolated from non-TBI control mice (7 weeks old) were seeded onto 96-multiwell culture plates (2 × 10^3^ cells/well) in growth medium supplemented without and with various concentrations (0–500 μM) of ferulic acid. After 48 h of incubation, the proliferation rate of BMSCs was assessed using Cell Counting Kit-8 (CCK-8; Dojindo Lab, Rockville, MD, USA) according to the manufacturer’s instructions. In addition, BMSCs were isolated from mice 30 and 60 days after TBI and cultured in 96-multiwell culture plates (2 × 10^3^ cells/well) containing growth medium. After 1, 3, and 5 days of incubation, proliferation rate of the cells was determined by CCK-8 assay. Optical density specific to the CCK-8 dye was measured at 450 nm using a microplate reader (SPECTROstar^®^ Nano, BMG LABTECH, Ortenberg, Germany).

### 2.8. Osteogenic and Adipogenic Differentiation Assays

BMSCs (5 × 10^4^ cells/well) isolated from mice at 30 or 60 days post-TBI were divided into 48-well culture plates in an osteogenic medium (αMEM supplemented with 100 nM dexamethasone, 50 μM ascorbic acid, 10 mM β-glycerol phosphate, and 5% FBS). During incubation, the medium was newly replaced to the same osteogenic medium every 2 days. After 21 days of incubation, cells were stained with 2% Alizarin red S (ARS, pH 4.2) for 20 min and observed under a light microscope. The stained cells were also treated with 10% cetylpyridinium chloride dissolved in 10 mM sodium phosphate (pH 7.0), and absorbance of the dye was measured at 405 nm using a microplate reader (SPECTROstar^®^ Nano). Parts of BMSCs (5 × 10^4^ cells/well) were seeded onto 48-well culture plates and incubated in an adipogenic medium (DMEM supplemented with 10 μg/mL insulin, 50 μM indomethacin, 1 μM dexamethasone, 500 μM 3-isobutyl-1-methylxanthine, and 5% FBS). After 21 days of incubation, cells were stained with 60% isopropanol containing 0.6% Oil red O (ORO), followed by capture of photographs. In addition, the ORO-stained cells were washed with water and treated with 100% isopropanol. The red-colored lipid droplets in these cells were quantified by determining the absorbance of the dye at 510 nm using a microplate reader (SPECTROstar^®^ Nano).

### 2.9. Quantitative Reverse Transcription-PCR (qRT-PCR) Assay

Total RNA was extracted from BMSCs that were isolated from mice at 30 and 60 days post-TBI using a TRIzol reagent (Invitrogen). Total RNA samples (1 μg/sample) were applied for cDNA synthesis using AmpiGene cDNA synthesis Kit (Enzo Life Sciences, Inc.) following the manufacturer’s instructions. The qRT-PCR was performed with Power SYBR Green PCR Master Mix (Applied Biosystems, Foster City, CA, USA) and ABI StepOnePlus Real-Time PCR System (Applied Biosystems). The thermocycling conditions were maintained at 95 °C for 10 min for pre-denaturation and amplified using three-step cycles of denaturation at 95 °C for 15 s, annealing at 60 °C for 30 s, and extension at 72 °C for 30 s for 40 cycles. Oligonucleotide primers specific to CCAA-enhancer-binding protein α (*C/EBPα*), peroxisome proliferator-activated receptor γ (*PPARγ*), and adiponectin (*apM1*) were designed as listed in Appendix A. The level of glyceraldehyde 3-phosphate dehydrogenase (*Gapdh*) was considered as the endogenous reference during the quantification.

### 2.10. Western Blot Analysis

Whole BM cells were isolated from femurs and tibias of mice at 7 days post-TBI. After removal of RBC, BM cells were lysed in a cocktail buffer containing protease/phosphatase inhibitor (Cell Signaling Technology, Danvers, MA, USA). The protein extracts (20 µg/sample) were separated through sodium dodecyl sulfate-polyacrylamide gel electrophoresis on 10–12% gels and electroblotted onto polyvinylidene difluoride membranes. Blots were washed with a buffer containing 10 mM Tris-HCl (pH 7.6), 150 mM NaCl, and 0.05% Tween-20, and blocked in 5% skim milk for 1 h prior to incubation with primary antibodies specific to PPARγ (1:500; SC-390740, Santa Cruz Biotechnology), adiponectin (1:500; ab22554, Abcam), and β-actin (1:2500; Santa Cruz Biotechnology). Finally, immunoreactive bands on the membranes were visualized using Western pico-EPD blot detection kit (ELPIS-Biotech, Daejeon, Republic of Korea), followed by exposure to X-ray film (Eastman Kodak, Rochester, NY, USA).

### 2.11. Assays for Enzyme Activities in Blood Serum and Liver Tissue

After 2 and 30 days of TBI, peripheral blood was isolated from mice by cardiac puncture and collected into serum separation tubes (BD Biosciences). Blood sera were obtained by centrifugating the tube at 10,000× *g* for 20 min, and then the activities of alanine amino transaminase (ALT) and aspartate aminotransferase (AST) were determined using specific assay kits (ADVIA 1650, Bayer, Japan). After collecting the blood sera, liver tissues were isolated from the mice and homogenized in 50 mM KH_2_PO_4_ solution for 5 min using a homogenizer (PRO Scientific Inc., Oxford, CT, USA), or in a 200-μL cold assay buffer provided by BioAssay Systems (Hayward, CA, USA). After centrifugation at 14,000× *g* for 10 min, the supernatants were collected, and activities of superoxide dismutase (SOD), catalase (CAT), and glutathione peroxidase (GPx) were determined. In these assays, SOD activity was measured using an assay kit (No. 706002, Cayman Chemical, Ann Arbor, MI, USA), while CAT and GPx activities were determined using an EnzyChrom™ CAT assay kit (ECAT-100; BioAssay Systems) and GPx assay kit (EGPX-100; BioAssay Systems), respectively, according to the manufacturer’s instructions.

### 2.12. Statistical Analyses

All results are expressed as the mean ± standard deviation (SD). One-way analysis of variance with post-hoc Tukey test was used for multiple comparisons using GraphPad Prism 8 (GraphPad Software Inc., San Diego, CA, USA). Two-tailed Student’s *t*-test was used when the significance of differences between two sets of data was determined using GraphPad Prism 8. A value of *p* < 0.05 was considered statistically significant.

## 3. Results

### 3.1. Supplemental Ferulic Acid Inhibits TBI-Mediated Impairments in Growth and Survival and Ameliorates HSC Senescence and Hematopoietic Defects in TBI Mice

A schematic diagram of the experimental designs, along with the chemical structure of ferulic acid, is shown in Figure 1A. All mice groups exhibited age-related growth during the experimental period, whereas the TBI group revealed significantly lower body weight from 30 or 45 days after TBI compared with the control or FA+TBI group (Figure 1B). No different body weights between control and FA+TBI groups were found throughout the periods. Compared with the FA+TBI group that showed 90% survival rate until 16 months, all mice in the TBI group died at the same month after TBI (Figure 1C). Survival rate of the FA+TBI group was similar to that of the control group, even at 20 months post-TBI (data not shown). As HSC senescence is a characteristic phenotype occurring under ionizing irradiation and contributes to hematopoietic disorders [16], we measured the SA-β-gal activity of HSCs (phenotypically defined by CD150^+^CD48^-^lineage-Sca-1^+^c-Kit^-^ cells) by determining the levels of C_12_FDG-positive HSCs (%) 2 and 30 days after TBI. The TBI group exhibited 2.97- and 1.98-fold higher levels of C_12_FDG-positive HSCs at 2 days post-TBI compared with control and FA+TBI groups, respectively (Figure 1D). The TBI group also showed higher SA-β-gal activity in HSCs than the control or FA+TBI group 30 days after TBI (Appendix A). However, the number of BM-conserved HSCs in the TBI group was comparable to those of the control and FA+TBI groups, both 2 and 30 days after TBI (Appendix A). We next determined levels of circulating blood cells in mice 2, 7, 30, and 60 days after TBI. Compared with the control group, numbers of circulating WBC, platelets, and RBCs, as well as percentage of granulocytes and lymphocytes in peripheral blood, were reduced in the TBI group in relation to the days after TBI (Figure 1E). TBI-mediated decreases in WBC, granulocytes, and lymphocytes were further visible at relatively early times compared with that in platelets and RBC. Treatment with ferulic acid blocked the TBI-induced reduction in circulating WBC and granulocytes, and these cells in the FA+TBI group were restored up to the levels of the control group 30 days after TBI. Furthermore, levels of lymphocytes, platelets, and RBC in FA+TBI mice were comparable with those in control mice throughout the days after TBI. These findings indicate that ferulic acid protects mice from TBI-induced defects in growth and survival, and this protection is, in part, associated with its potential to inhibit HSC senescence and acute hematopoietic defects.

### 3.2. Treatment with Ferulic Acid Inhibits ROS Accumulation and Senescence Induction in MSCs of TBI Mice

While MSCs play critical roles in bone mass accrual, irradiation can induce senescence of these cells via ROS-activated signaling [17]. We determined mitochondrial ROS levels and SA-β-gal activity in MSCs (phenotypically defined by Lin^−^, Sca-1^+^, c-Kit^+^, CD105^+^, and CD29^+^) of mice 2, 30, and 60 days after TBI. MSCs from the TBI group exhibited an obvious increase in mitochondrial ROS level compared with cells from control mice, and this increase was significantly diminished by intraperitoneal injection of ferulic acid (Figure 2A,B). Similar to HSCs, the TBI group revealed greater SA-β-gal activity in MSCs compared with cells from control mice, while the SA-β-gal activity in MSCs from FA+TBI mice was comparable with that of control mice every day post-TBI (Figure 2C,D). The number of MSCs in BM of TBI and FA+TBI groups was also similar to that of control mice 2, 7, and 30 days after TBI (data not shown). These findings support the antioxidant and antisenescence effects of ferulic acid on BM-conserved MSCs in TBI mice.

### 3.3. Supplementation with Ferulic Acid Limits TBI-Induced Defect of BM Microenvironment

The BM microenvironment provides niches for retention and self-renewal of BM HSCs and MSCs, and its impairment is related to abnormal retention and senescence of these cells, along with bone mass loss [18]. We explored whether the TBI-induced stem cell senescence and body weight loss is associated with impaired BM microenvironments. The 2D μCT analysis 60 days after TBI showed greater bone mass in control and FA+TBI groups compared with the TBI group (Figure 3A). When values of bone parameters were evaluated, the TBI group revealed significantly lower values of BV (*p* < 0.01), BS (*p* < 0.01), BV/TV (*p* < 0.01), and BMD (*p* < 0.05) compared with the control group (Figure 3B). TBI-induced decreases in the bone values were restored in mice supplemented with ferulic acid up to the levels of control mice. Ferulic-acid-induced protection on TBI-mediated bone loss was supported by H&E staining, in which the TBI group showed relatively lower bone mass in the trabecular region at 60 days post-TBI than the control or FA+TBI group (Figure 3C).

### 3.4. Administration of Ferulic Acid Increases Osteogenic Marker Expression, but Inhibits Osteoclastic Activation in BM of TBI Mice

As a balanced activation between osteoblasts and osteoclasts is important in maintaining the BM microenvironment and bone mass accrual, we explored whether TBI-mediated BM impairment is directly associated with an alteration in osteoblastic and osteoclastic activation. Compared with the control group, the TBI group revealed significantly lower levels of RUNX2 (*p* < 0.05) and OCN (*p* < 0.01) in BM 60 days after TBI (Figure 4). The TBI-mediated decreases in these factors were completely recovered up to the levels of the control group by treatment with ferulic acid. In contrast, the TBI group exhibited significantly higher levels (*p* < 0.001) of RANKL and CTSK in BM compared with control and FA+TBI groups (Figure 4B). TBI groups also exhibited a significantly lower level (*p* < 0.05) of OPG in BM compared with the control group, whereas this reduction was not affected by administration of ferulic acid (Figure 4C). Alternatively, the TBI group showed significantly higher numbers of TRAP-positive osteoclasts than did the control (*p* < 0.01) or FA+TBI group (*p* < 0.05) (Figure 4D). These results not only indicate a relationship between TBI-mediated BM injury and a preferable activation toward osteoclasts with reduced osteogenic activity, but also suggest that ferulic acid inhibits osteoclastic activation by decreasing the induction of RANKL and CTSK, rather than by increasing OPG in BM.

### 3.5. Ferulic Acid Inhibits TBI-Stimulated Adipogenic Differentiation in BM

An adipogenic activation in BM is one of the important alterations after ionizing irradiation, and this contributes to deterioration in bone formation [19]. We assessed whether TBI actually stimulates adipogenic differentiation in BM, and whether this is also suppressed by supplementation with ferulic acid. To address this, we initially checked the presence of lipid spot formation in BM of mice and counted its numbers 60 days after TBI (Figure 5A). As shown in the images of H&E staining, the numbers of lipid spots in BM of the TBI group were significantly higher compared with that of the control (*p* < 0.001) or FA+TBI group (*p* < 0.001) at 60 days post-TBI. TBI-mediated increase in lipid accumulation was correlated with the expression levels of adipogenic markers, adiponectin and PPARγ, in which supplemental ferulic acid significantly (*p* < 0.05) diminished expression of these markers in the BM (Figure 5B). To further evaluate the impacts of TBI or in combination with ferulic acid on adipogenesis in BM, we determined levels of *PPARγ*, *C/EBPα*, and *apM1* in BM cells of mice by qRT-PCR assay 30 and 60 days after TBI. The TBI group exhibited significantly higher levels (*p* < 0.001) of the adipogenic marker genes in BM cells compared with the control group 30 days after TBI (Figure 5C). TBI-mediated increases in *PPARγ*, *C/EBPα*, and *apM1* in BM cells were also found 60 days after TBI (Figure 5D). Treatment with ferulic acid diminished TBI-mediated upregulation of the adipogenic regulatory genes at both 30 and 60 days post-TBI, in which levels of *PPARγ* and *C/EBPα* in the FA+TBI group were comparable with those in control mice 60 days after TBI. These results indicate that, in addition to osteoclastic activation, TBI augments adipogenic activation in BM and BM-conserved cells, and this augmentation is attenuated by supplemental ferulic acid.

### 3.6. Ferulic Acid Treatment Stimulates Proliferation and Osteogenic Activation, but Inhibits Adipogenic Differentiation of BMSCs from TBI Mice

We evaluated how a direct addition of ferulic acid affects proliferation of BMSCs from non-TBI control mice. When BMSCs were incubated in the presence of ferulic acid (0–500 μM) for 48 h, the cells exhibited a dose-dependent proliferation up to a ferulic acid concentration of 200 μM (Figure 6A). To further understand the impacts of supplemental ferulic acid on BMSC proliferation and their differentiation into osteoblastic or adipocytic lineage cells, BMSCs isolated from mice at various days post-TBI were incubated in growth or differentiating medium. When BMSCs from mice 30 (Figure 6B) and 60 days after TBI (Figure 6C) were incubated in growth medium for 1, 3, or 5 days, cells from all mice groups showed a time-dependent increase in optical density specific to CCK-8. However, BMSCs from the TBI group exhibited a significantly lower proliferation rate compared with the cells from the control or FA+TBI group at the same incubation times. No different values of optical density between the control and FA+TBI groups were found throughout the incubation. TBI-mice-derived BMSCs exerted relatively lower intensity specific to ARS compared with cells from the control or FA+TBI group, and this was further viable in the cells isolated 60 days rather than 30 days after TBI (Figure 6D). Determination of the mean optical density specific to the ARS dye also supported TBI-mediated decrease in mineralization of BMSCs, and its complete suppression by supplementation with ferulic acid, both at 30 and 60 days post-TBI (Figure 6E). When BMSCs isolated from mice at 30 days post-TBI were incubated in adipogenic medium for 21 days, TBI-mice-derived cells showed greater levels of ORO-positive adipocytes along with higher ORO-specific optical density compared with cells from the control or FA+TBI group (Figure 6F). Western blot analysis also showed greater immunoreactive intensities of PPARγ and adiponectin in BMSCs from the TBI group compared with cells from the control or FA+TBI group 7 days after TBI (Figure 6G, left panel). TBI-mediated increase of PPARγ in BMSCs and its inhibition by ferulic acid were further supported by measuring the mean immunoreactive intensities of PPARγ and adiponectin at the same day post-TBI (Figure 6G, right panel).

### 3.7. Supplementation with Ferulic Acid Inhibits TBI-Mediated Increases in AST and ALT Activities, but Enhances Cellular Antioxidant Defense Systems

As TBI is known to acutely increase activities of AST and ALT, which are indicative of liver damage [9], we determined activities of the enzymes in blood serum 2 and 30 days after TBI. The TBI group showed significantly higher activity (*p* < 0.05) of AST at 2 days, but not at 30 days post-TBI compared with the control or FA+TBI group (Figure 7A). Similarly, ALT activity in the TBI group was significantly higher (*p* < 0.05) than that in the control or FA+TBI group at 2 days post-TBI (Figure 7B). Different to the liver-damage-indicative enzymes, the activities of liver-conserved antioxidant enzymes, such as SOD (Figure 7C), CAT (Figure 7D), and GPx (Figure 7E), in the TBI group were acutely diminished 2 days, but not 30 days, after TBI. Treatment with ferulic acid completely blocked TBI-mediated decreases in activities of the enzymes at 2 days post-TBI. In particular, the FA+TBI group revealed significantly higher SOD activity (*p* < 0.01) than the control or TBI group at 30 days post-TBI (Figure 7C). Furthermore, IHC assay revealed that the FA+TBI group expressed an approximately 2.5-fold higher level (*p* < 0.01) of NRF2 in BM at 60 days post-TBI compared with that in the control or TBI group (Figure 7F). These results suggest that long-term supplementation with ferulic acid enhances antioxidant defense systems, specifically SOD activity and NRF2 induction, and this enhancement is closely associated with ferulic-acid-induced protection on TBI-mediated oxidative damages.

## 4. Discussion

Although therapeutic radiation can cause severe damages to intact soft and hard tissues [20,21], TBI is required for cancer patients or in BM transplantation. Reduced bone mass accrual and BMD, increased ROS and oxidative stress, long-term residual BM injury, and stem cell senescence with the attendant hematopoietic defects are to be the hallmarks of TBI-mediated disorders [9,22,23,24]. Disruption of cellular redox balance and depletion of endogenous antioxidant defense systems are also the main events in TBI-mediated oxidative damages [9,25]. As cellular ROS accumulation and subsequent oxidative stress to cells and tissues are the key mediators in TBI-mediated impairments, supplemental antioxidants may ameliorate or recover TBI-mediated oxidative defects. Here, we highlight a long-lasting radioprotective potential of ferulic acid in mice exposed to a sub-lethal TBI. This study also provides the underlying mechanisms by which long-term administration of the antioxidant limits stem cell senescence and BM injuries, and encourages survival and bone mass accrual in TBI mice.

We previously found that, in addition to ferulic acid, dietary antioxidants, caffeic acid and coumaric acid prevent TBI-mediated damages of liver and spleen tissues and HSC senescence, as well as restore activities of endogenous antioxidant enzymes [9]. Caffeic acid also exhibited an autonomous ameliorating effect on HSC-senescence-associated long-term BM injury and mortality in TBI mice, and induced in vitro anticancer activity, thereby indicating its clinical usefulness as a supplemental drug for cancer patients [24]. Different to the previous studies [9,24], this study administered mice with ferulic acid via intraperitoneal injection once per day for 37 consecutive days (7 days before and 30 days after TBI), and TBI-associated phenotypes in mice were analyzed at various time points after TBI. These experimental approaches allow the impacts of radioprotective antioxidant on TBI-mediated systemic damages to be defined in regard to its long-term administration, as well as to evaluate a time-dependent alteration of TBI-associated characteristic phenotypes. Our results support that intraperitoneal injection of ferulic acid inhibits TBI-mediated HSC senescence and acute reduction of circulating blood cells. As TBI-mediated defects in hematopoietic development may increase the risk of bleeding, infection, and mortality, our findings also indicate that long-term administration of ferulic acid protects mice against TBI-mediated mortality by restoring levels of circulating blood cells.

TBI-mediated defects in body growth and bone mass accrual can be associated with BM microenvironmental impairments that cause functional loss or dysregulated differentiation of BM-conserved stem cells. Indeed, exposure to TBI induced a long-term residual BM injury, depending on the exposed dose or time of TBI, and this negatively affected BM niches and bone homeostasis [16,24]. In addition to HSCs, TBI induced oxidative DNA damage of MSCs and defected their functionality [6,26,27]. TBI also impaired proliferation, differentiation, and chromosomal integrity in BMSCs [28]. Together, our results with previous reports suggest that the ferulic-acid-induced recovery of body growth and bone mass accrual in TBI mice is, in part, associated with its potential to limit ROS accumulation in MSCs and their senescence induction, as well as to maintain functionality of the cells.

The multipotency of MSCs are dysregulated after TBI, in which a preferable differentiation toward adipocytes, along with an imbalanced activation between osteogenic and osteoclastic differentiation, occurs in BM [7]. TBI diminishes proliferation of osteoblastic progenitor cells with cell cycle arrest, and decreases the production of bone-consisting components [29]. ROS accumulation and subsequent oxidative stress are also closely associated with osteoclastogenesis, skeletal aging, and bone diseases [30]. BM adipocytes account for approximately 10% of total body fat in healthy adults and play important roles in energy storage, endocrine function, and bone metabolism [31]. However, chemotherapy or irradiation can cause abnormal infiltration and accumulation of adipocytes in BM [7]. As proven by the decreased induction of RUNX2, OCN, and OPG, upregulated RANKL and CTSK induction, along with increased osteoclasts, and increases in lipid accumulation and adipogenic marker expression in BM of mice 60 days after TBI, the current findings suggest that TBI-mediated BM injury is closely associated with the dysregulated differentiation or functions of HSC- or MSC-derived progenitor cells in BM. Our findings also support that long-term administration of ferulic acid mostly suppresses TBI-mediated osteoclastic activation and restores osteoblastic activity in BM by activating osteogenic markers and by inhibiting directly osteoclastic activation, rather than by increasing OPG expression. Additionally, this study demonstrates that the inhibitory effect of ferulic acid on TBI-mediated adipogenic activation is also accompanied by its ability to diminish the expression of *PPARγ*, *C/EBPα*, and *apM1*, which are known to differentiate preadipocytes into adipocytes and to inhibit osteoblastic differentiation [32]. It was reported that ferulic acid attenuates adipocyte differentiation in 3T3-L1 cells via a positive regulation of heme oxygenase-1, a downstream effector of NRF2 [33]. Adipogenic activation in BM is also associated with a delayed engraftment of HSCs and hematopoietic defects [19]. Therefore, it is likely that supplemental ferulic acid inhibited TBI-stimulated adipogenesis in BM by activating NRF2-related antioxidant systems and by preventing senescence of HSCs and oxidative BM injury. All results from BMSC cultures also support that supplemental ferulic acid increases proliferation and differentiation of BMSCs into mineralized cells, but suppresses their differentiation into adipocytes.

Endogenous antioxidant defense systems maintain cellular redox balance and protect against acute or prolonged oxidative stress, and, thus, a defect in the systems facilitates ROS accumulation and systemic oxidative damages. Similar to the previous study [9], the present findings showed that TBI acutely induces liver injury and depletes the activities of SOD, CAT, and GPx, whereas these defects are recovered by supplemental ferulic acid. In particular, the activity of SOD, but not of CAT and GPx, in the FA+TBI group was greater than that in control mice, thereby indicating an enhancing effect of ferulic acid on SOD. Enzymes such as SOD, CAT, and GPx are the primary antioxidant defense systems that convert active oxygen molecules into nontoxic substances and/or remove directly reactive oxidants [34,35]. NRF2 is a sensitive sensor in response to cellular metabolic and stress states and plays a role as a key transcriptional regulator for antioxidant enzymes [36]. When exposed to oxidative stress, NRF2 is translocated into the nucleus and selectively activates gene transcription corresponding to various endogenous antioxidant enzymes, including CAT and SOD [37]. This indicates that upregulation of NRF2 ameliorates TBI-mediated DNA damages by enhancing the antioxidant defense system or by maintaining an intracellular redox balance [36,38,39]. Activation of NRF2 mitigated TBI-mediated hematopoietic death and augmented function and regeneration of HSCs, thereby increasing survival in TBI mice [40,41,42]. Taken as a whole, it is suggested that NRF2 is a potential therapeutic target to ameliorate or prevent TBI-mediated oxidative injuries, as well as to regulate self-renewal, proliferation, and differentiation of stem cells [43]. Overall, our current findings with previous reports [9,24] strongly suggest that the ferulic-acid-induced radioprotection is closely associated with the enhanced or maintained antioxidant defense systems that are followed by NRF2 upregulation.

## 5. Conclusions

In summary, small molecule antioxidants, such as dietary flavonoids and phenolic acids, may enhance HSC function and improve the efficacy of HSC transplantation therapy via the NRF2-related maintenance of BM and BM-conserved stem cells [42]. This study demonstrates that supplemental ferulic acid protects mice from TBI-mediated BM injuries and senescence of BM-conserved HSCs and MSCs by inhibiting ROS accumulation and osteoclastic and adipocytic activation in BM, as well as by recovering osteogenic activity and NRF2-associated antioxidant defense systems. This study also highlights that long-term administration of ferulic acid improves bone mass accrual and survival, and specifically accelerates SOD activity and NRF2 induction. The current findings indicate a preclinical potential of ferulic acid to prevent TBI-mediated oxidative disorders and to enhance therapeutic efficacy in BM transplantation. Overall, our results of this study imply that dietary phenolic antioxidants are attractive candidates as injectable materials for radioprotection.

## Figures and Tables

**Figure 1 antioxidants-10-01209-f001:**
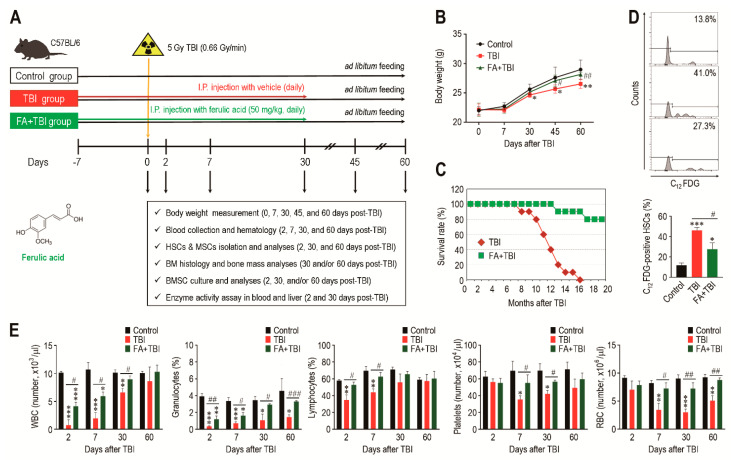
Supplemental ferulic acid diminishes HSC senescence in BM of TBI mice and restores TBI-mediated defects in body growth, survival, and production of circulating blood cells. (**A**) The experimental designs and chemical structure of ferulic acid. (**B**) Body weight (g) of mice groups were measured at the indicated days after TBI (*n* = 10). (**C**) Survival rate of mice groups was monitored for the indicated months (*n* = 10). (**D**) SA-β-gal activity in HSCs of mice groups was determined by measuring percentage of C_12_FDG-positive HSCs at 2 days post-TBI via flow cytometric analysis (*n* = 4). (**E**) Levels of circulating WBC, granulocytes, lymphocytes, platelets, and RBC in mice groups were measured using an automated complete blood cell counter at the indicated days after TBI (*n* ≥ 3). All data are presented as the mean ± SD. * *p* < 0.05, ** *p* < 0.01, and *** *p* < 0.001 compared with control group; ^#^ *p* < 0.05, ^##^ *p* < 0.01, and ^###^ *p* < 0.001 compared with TBI group.

**Figure 2 antioxidants-10-01209-f002:**
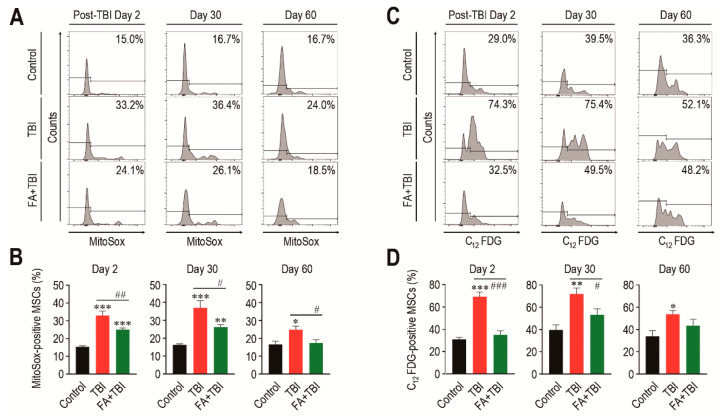
Supplemental ferulic acid inhibits TBI-mediated increases of ROS accumulation and senescence induction in BM-conserved MSCs. (**A**,**B**) Level of mitochondrial superoxide anions in BM MSCs from mice groups were assessed by flow cytometry using MitoSox^TM^ Red reagent at the indicated days after TBI (*n* = 4). (**C**,**D**) SA-β-gal activity in BM-conserved MSCs from mice groups was also measured by flow cytometry using C_12_FDG at the same post-TBI days (*n* = 4). All data are presented as the mean ± SD. * *p* < 0.05, ** *p* < 0.01, and *** *p* < 0.001 compared with control group; ^#^ *p* < 0.05, ^##^ *p* < 0.01, and ^###^ *p* < 0.001 compared with TBI group.

**Figure 3 antioxidants-10-01209-f003:**
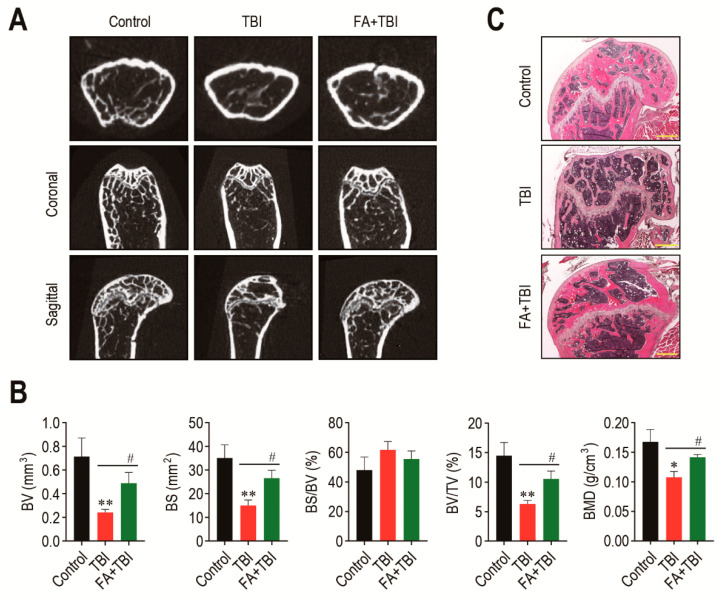
Administration of ferulic acid restores TBI-mediated impairments in BM microenvironment and bone mass accrual. (**A**) The 2D μCT images show femoral bones of mice groups 60 days after TBI. (**B**) Values of bone parameters, including BV (mm^3^), BS (mm^2^), BS/BV (%), BV/TV (%), and BMD (g/cm^3^), in the trabecular region of the femoral bones were measured 60 days after TBI (*n* = 4). (**C**) Histological evaluation of bone mass accrual at the trabecular bones of mice groups was performed by H&E staining at 60 days post-TBI. A representative result from four different samples is shown. Scale bar = 500 μm. All data are presented as the mean ± SD. * *p* < 0.05 and ** *p* < 0.01 compared with control group; ^#^ *p* < 0.05 compared with TBI group.

**Figure 4 antioxidants-10-01209-f004:**
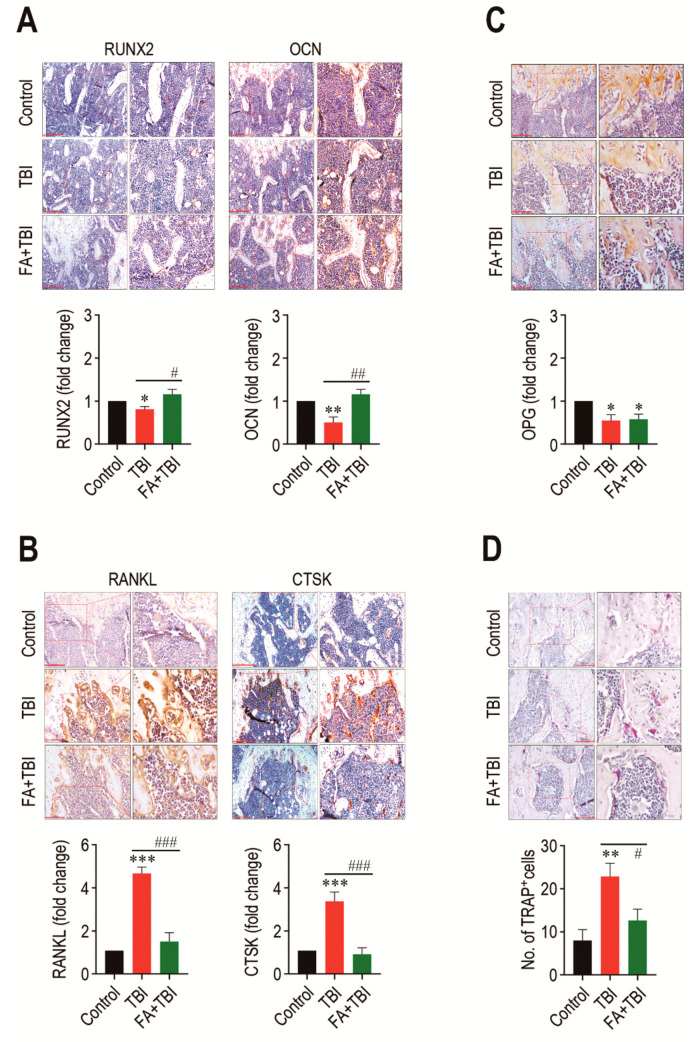
TBI causes an imbalanced activation between osteoblasts and osteoclasts, and this is completely prevented by long-term administration of ferulic acid. Levels of (**A**) RUNX2 and OCN, (**B**) RANKL and CTSK, and (**C**) OPG in BM of mice groups were evaluated by IHC assay at 60 days post-TBI. (**D**) Osteoclasts formed in BM of mice were determined by TRAP staining at the same day after TBI. The IHC images show representative results from four different mice. Scale bars in the images for RUNX2, OCN, and CTSK are 200 μm, while those for RANKL, OPG, and TRAP are 100 μm. All data are presented as the mean ± SD (*n* = 4). * *p* < 0.05, ** *p* < 0.01, and *** *p* < 0.001 compared with control group; ^#^ *p* < 0.05, ^##^ *p* < 0.01, and ^###^ *p* < 0.001 compared with TBI group.

**Figure 5 antioxidants-10-01209-f005:**
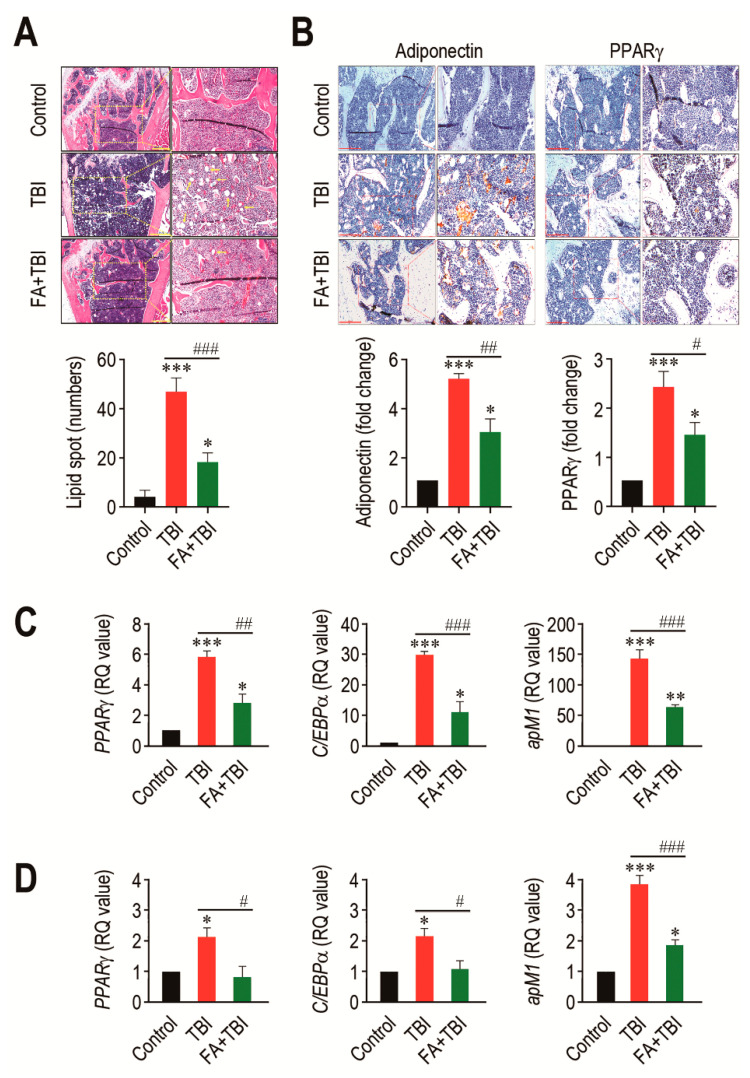
Long-term administration of ferulic acid inhibits TBI-mediated adipogenic activation in BM of mice. (**A**) Lipid accumulation in trabecular regions of mice groups was determined by H&E staining at 60 days post-TBI. Yellow arrows indicate the lipid spots formed in the bone. Scale bar = 500 μm. (**B**) Levels of adiponectin and PPARγ in BM of mice groups were measured by IHC assay at 60 days post-TBI. Scale bar = 200 μm. Number of lipid spots and relative intensity of adiponectin and PPARγ (fold change) in BM were calculated from four different samples. Expression levels of *PPARγ*, *C/EBPα*, and *apM1* in BM cells from mice groups were determined by qRT-PCR assay at (**C**) 30 and (**D**) 60 days post-TBI (*n* = 4). All data are expressed as the mean ± SD. * *p* < 0.05, ** *p* < 0.01, and *** *p* < 0.001 compared with control group; ^#^ *p* < 0.05, ^##^ *p* < 0.01, and ^###^ *p* < 0.001 compared with TBI group.

**Figure 6 antioxidants-10-01209-f006:**
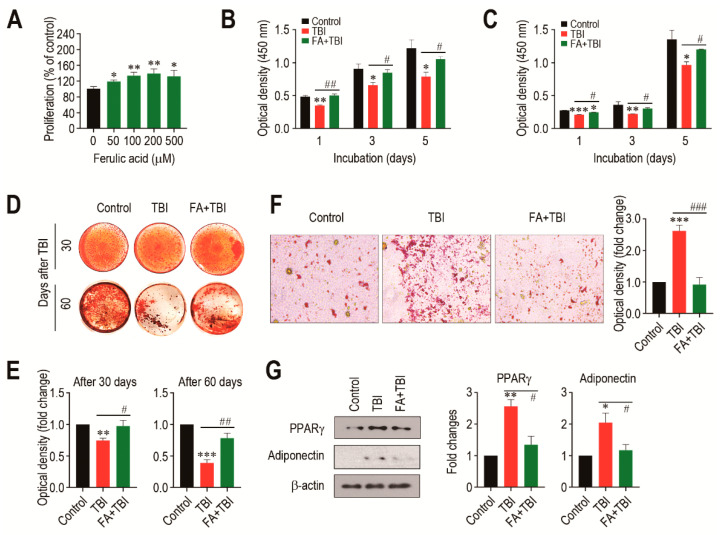
Supplemental ferulic acid restores proliferation and osteogenic differentiation of BMSCs and prevents adipogenic activation of the cells from TBI mice. (**A**) BMSCs isolated from non-TBI control mice were incubated in growth medium supplemented with ferulic acid (0–500 μm) for 48 h, and proliferation rate was determined by CCK-8 assay. BMSCs were isolated from mice exposed to TBI or in combination with ferulic acid at (**B**) 30 and (**C**) 60 days post-TBI, and the cells were incubated in growth medium for the indicated days, followed by CCK-8 assay. (**D**) BMSCs were isolated from mice groups at 30 and 60 days post-TBI and incubated in osteogenic medium. After 21 days of incubation, cellular mineralization of the cells was evaluated by ARS staining. (**E**) Mineralization of BMSCs from mice groups was also determined by measuring optical density specific to the ARS at 405 nm. (**F**) BMSCs were isolated from mice groups 30 days after TBI and incubated in adipogenic medium. The photographs showing ORO-positive adipocytes and optical density specific to the dye were obtained 21 days after incubation. (**G**) BM cells were isolated from mice groups 7 days after TBI and, after 48 h of incubation in adipogenic medium, protein levels of PPARγ and adiponectin in the cells were determined by immunoblot assay. The relative intensities specific to PPARγ and adiponectin were determined after normalizing the band to the relative intensity of β-actin. All data are presented as the mean ± SD (*n* ≥ 3). * *p* < 0.05, ** *p* < 0.01, and *** *p* < 0.001 compared with control group; ^#^ *p* < 0.05, ^##^ *p* < 0.01, and ^###^ *p* < 0.001 compared with TBI group.

**Figure 7 antioxidants-10-01209-f007:**
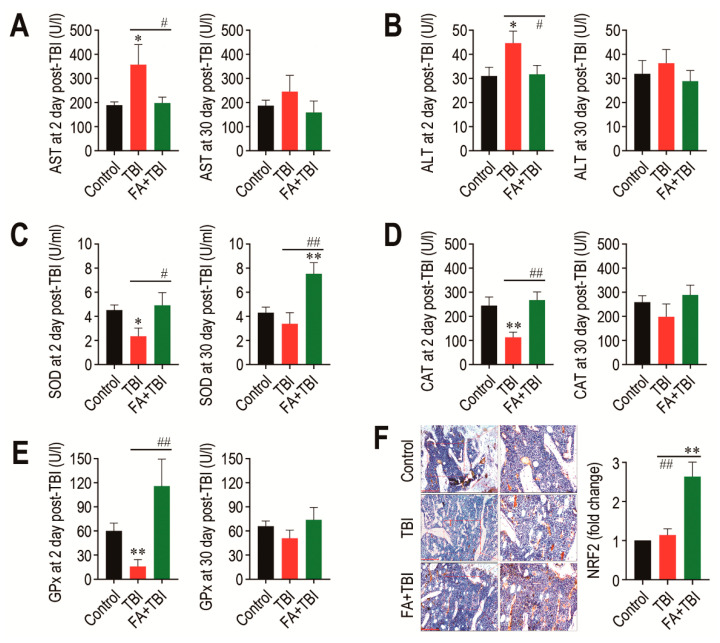
Long-term supplementation with ferulic acid prevents liver damage and increases the activities of liver-conserved antioxidant enzymes and the induction of NRF2 in BM of TBI mice. Levels of (**A**) AST and (**B**) ALT in blood serum of mice groups were determined 2 and 30 days after TBI. The activities of (**C**) SOD, (**D**) CAT, and (**E**) GPx in the liver from mice groups were determined 2 and 30 days after TBI. (**F**) Level of NRF2 in BM of mice groups was evaluated by IHC assay at 60 days post-TBI. Scale bar = 200 μm. All data are presented as the mean ± SD (*n* = 4). * *p* < 0.05 and ** *p* < 0.01 compared with control group; ^#^ *p* < 0.05 and ^##^ *p* < 0.01 compared with TBI group.

## Data Availability

Data is contained within the article and Appendix A.

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
