# Peer review of "Supplemental Ferulic Acid Inhibits Total Body Irradiation-Mediated Bone Marrow Damage, Bone Mass Loss, Stem Cell Senescence, and Hematopoietic Defect in Mice by Enhancing Antioxidant Defense Systems"

_antioxidants, 2021, doi:10.3390/antiox10081209_

Round 1

Reviewer 1 Report

Wagle et al present a well-designed and thorough investigation on the effects of the antioxidant Ferulic Acid on the irradiated bone marrow microenvironment. I have only a few minor comments/corrections to lend to an overall well-written and interesting manuscript.

Line 83. Please remove or qualify "without adverse effects". You haven't shown conclusively that this is the case in the current study. While your tumour-naive mouse data may suggest this, it's questionable whether this treatment wouldn't exacerbate hemopoietic cancers. That said, I can certainly see a role for this kind of treatment prior to and post-BM transplantation.

Line 86. Please indicate CAS number and whether the ferulic acid is a mixture of isomers or not.

Line 101. Please indicate the vehicle used.

Line 244. Please indicate whether one- or two-tailed tests were used.

Line 258. It would be helpful to the reader to indicate the surface markers used to select HSCs in this section of the text.

Figure 3. Do the authors have any idea what the minimal dose might be to achieve these outcomes? Did they try doses lower than 50mg/kg/day? How did they arrive at this dosage?

Author Response

Author's Reply to the Review Report (Reviewer 1)

English language and style

(x) English language and style are fine/minor spell check required

►Author response: The manuscript was re-checked by a native English-speaking colleague before submission.

Comments and Suggestions for Authors

Wagle et al present a well-designed and thorough investigation on the effects of the antioxidant Ferulic Acid on the irradiated bone marrow microenvironment. I have only a few minor comments/corrections to lend to an overall well-written and interesting manuscript.

Line 83. Please remove or qualify "without adverse effects". You haven't shown conclusively that this is the case in the current study. While your tumour-naive mouse data may suggest this, it's questionable whether this treatment wouldn't exacerbate hemopoietic cancers. That said, I can certainly see a role for this kind of treatment prior to and post-BM transplantation.

►Author response: Thank you for the helpful comments. As the reviewer mentioned, there were no data to directly support our statement “without adverse effects”. In the revised manuscript, we deleted the statement. We also deleted this statement in the Conclusions section.

Line 86. Please indicate CAS number and whether the ferulic acid is a mixture of isomers or not.

►Author response: The authors indicated the CAS number of ferulic acid (CAS:537-98-4) and the chemical is trans-ferulic acid with 99% purity.

Line 101. Please indicate the vehicle used.

►Author response: In this study, we used phosphate-buffered saline as the vehicle solution. We indicated this within the text. 

Line 244. Please indicate whether one- or two-tailed tests were used.

►Author response: The authors indicated that two-tailed Student’s t-test was used in this manuscript.

Line 258. It would be helpful to the reader to indicate the surface markers used to select HSCs in this section of the text.

►Author response: Although the surface markers to define HSCs were described in the materials and methods section (section 2.3.), we briefly indicated the HSC surface markers utilized in the Results section.

Figure 3. Do the authors have any idea what the minimal dose might be to achieve these outcomes? Did they try doses lower than 50mg/kg/day? How did they arrive at this dosage?

►Author response: Thank you for your comments. The dose of ferulic acid for animal studies was determined on the bases of previously published studies; Das et al., Plos One. 2014, 9, e97599, doi:10.1371/jour-nal.pone.0097599 (Ref. No. 12 in this manuscript); Das et al., Free Radic. Res., DOI: 10.1080/10715762.2016.1267345; Ma et al., Int. J. Radiat. Biol., 2011, 87, 499–505; Zhang et al., Int. J. Immunopatho. Pharmacol. 2018, 31:1-9. Especially, we selected the dose (50 mg/kg), because that the dose was the minimum dosage to protect completely survival of mice exposed to lethal TBI (10 Gy) at one month post-TBI [Das et al., 2014;9:e97599, Plos One]. When mice were administered with the dose (25 mg/kg) of ferulic acid, the survival rate was apparently reduced from 19 days after TBI, indicating the dose with 50 mg/kg is the minimal does to achieve the ferulic acid-derived outcomes.

Reviewer 2 Report

It is an interesting data and suggestions. There are minor comments on the manuscript.

a. Please, add bar graph description on figure 1E (indication of red and black bar). 

b. It would be nice to add mechanism study about PPAR-gamma and other protein connection through western blotting analysis. 

c. Is it possible to bone marrow transplantation experiments? (between TBI and FA-TBI). In this case, FA supplemented bone marrow could secret in TBI bone marrow to inhibit ROS accumulation in the animal.

d. In the conclusion part, authors need to modify FA as a dietary supplement to injection material. Since, experimental design was i.p. injection not oral administration. 

Author Response

Author's Reply to the Review Report (Reviewer 2)

English language and style

(x) Moderate English changes required

►Author response: According to the reviewers’ comment, we tried to improve English editing via checking by a native English-speaking colleague before submission.

Comments and Suggestions for Authors

It is an interesting data and suggestions. There are minor comments on the manuscript.

  1. Please, add bar graph description on figure 1E (indication of red and black bar). 

►Author response: It was our mistake. We indicated the bar description within the figure.

  1. It would be nice to add mechanism study about PPAR-gamma and other protein connection through western blotting analysis. 

►Author response: Thank you for your comment. In addition to the PPARgamma, we added the immunoblot result of adiponection in Figure 6G.

c. Is it possible to bone marrow transplantation experiments? (between TBI and FA-TBI). In this case, FA supplemented bone marrow could secret in TBI bone marrow to inhibit ROS accumulation in the animal.

►Author response: Thank you for your helpful comments. Though we did not perform the BM transplantation experiments, we suggest that the administration of ferulic acid might ameliorate ROS stress, as well as suppress BM HSC senescence in the recipient mice that exposed to TBI. This suggestion was due to our previous findings showing that competitive transplantation of equal numbers of CD150+CD48-LSK cells from caffeic acid-supplemented mice with cells from competitor mice (CD45.1) into conditioned recipient mice (CD45.1/2) that were lethally irradiated protected greater survival of the recipients with higher donor cell-proliferation compared with the recipients transplanted with the cells from TBI alone (Sim et al., Aging and Disease, 2019, 10; 1320-1327, Ref. No. 24 in this manuscript).

  1. In the conclusion part, authors need to modify FA as a dietary supplement to injection material. Since, experimental design was i.p. injection not oral administration. 

►Author response: The authors agreed with the reviewers’ comment. We revised the sentence in the Conclusions.
